# Zinc Nutritional Status in a Series of Children with Chronic Diseases: A Cross-Sectional Study

**DOI:** 10.3390/nu13041121

**Published:** 2021-03-29

**Authors:** Marlene Fabiola Escobedo-Monge, María Carmen Torres-Hinojal, Enrique Barrado, María Antonieta Escobedo-Monge, José Manuel Marugán-Miguelsanz

**Affiliations:** 1Faculty of Medicine, Valladolid University, Avenida Ramón y Cajal, 7, 47005 Valladolid, Spain; mctorresh@telefonica.net; 2Department of Analytical Chemistry, Science Faculty, University of Valladolid, Campus Miguel Delibes, Calle Paseo de Belén, 7, 47011 Valladolid, Spain; ebarrado@qa.uva.es; 3Department of Chemistry, Science Faculty, University of Burgos, Plaza Misael Bañuelos sn, 09001 Burgos, Spain; antoitalia777@gmail.com; 4Department of Pediatrics of the Faculty of Medicine, Valladolid University, Section of Gastroenterology and Pediatric Nutrition, University Clinical Hospital of Valladolid, Avenida Ramón y Cajal, 7, 47005 Valladolid, Spain; jmmarugan@telefonica.net

**Keywords:** serum zinc level, dietary zinc intake, hypozincemia, dietary zinc deficiency, body mass index, iron, magnesium

## Abstract

Background: Zinc is an essential trace element for the normal growth and development of human beings. The main objective was to evaluate the nutritional status of zinc and its association with nutritional indicators in a series of children with chronic diseases. Methods: The prevalence of patients with dietary zinc deficiency or deficit zinc intake (<80% DRI: dietary reference intake) was analyzed through prospective 72 h dietary surveys, and serum zinc deficiency or hypozincemia (≤70 µg/dL in children under 10 years of age in both sexes and in females older than 10 years and <74 μg/dL in males older than 10 years) was measured through atomic absorption spectrophotometry. The participants were classified according to their nutritional status by body mass index (BMI). Results: Mean serum zinc level in obese (87 µg/dL), undernourished (85 µg/dL), and eutrophic children (88 µg/dL) were normal, but in the undernutrition (60% DRI) and eutrophic (67% DRI) groups the mean dietary zinc intake was low compared to that in the obesity group (81% DRI). There were different associations between nutritional parameters, dietary zinc intake, and serum zinc. All patients with hypozincemia had dietary zinc deficiency. Conclusions: In the whole series, 69% of participants showed a zinc intake lower than recommended and might be at high risk of zinc deficiency.

## 1. Introduction

Diseases can be divided into short-term and long-term condition [1]. By definition, a chronic disease is a long-term, non-curable [2]. There are diseases that, with a minor evolution over time, are from the beginning, by definition, chronic [3]. According to the World Health Organization (WHO), malnutrition refers to deficiencies, excesses, or imbalances of energy and/or nutrients in a person’s dietary intake [4]. Undernutrition has detrimental effects on organ development and growth [5], causing two of the best-known forms of undernutrition, stunting (chronic, intergenerational undernutrition) and wasting (acute undernutrition) [6]. Furthermore, childhood obesity and its comorbidities have become a major health problem worldwide [7]. Developing countries are undergoing a progressive rise in obesity and nutrition-related chronic diseases (NRCDs). In transitional countries, stunting and micronutrient deficiencies such as those of iron, vitamin A, and zinc coexist in children with obesity and NRCDs, creating a double burden of nutritional disease [8].

Zinc is one of the most common trace minerals in the human body and performs a significant role in growth and development, acting as a crucial structural, catalytic, and signaling factor [9] for proteins involved in cell proliferation, migration, survival, death, neuronal development, immunity, and signal transduction [10]. Zinc deficiency causes stunted growth, compromises the immune and nervous systems, affecting practically all other organ systems. The most common clinical manifestations in individuals are skin lesions, depression of mental functions, impaired night vision, anorexia, hypogonadism, delayed wound healing, and impaired immune functions. Moreover, zinc deficiency at the cellular level and its overload originate oxidative stress. Suitable control of cellular zinc is vital for the equilibrium between health and disease [11]. Clinical observations have established that zinc deficiency is related to pathologic changes in many diseases [12], and dysregulation of epigenetics due to its deficiency may be implied in the pathogenesis of the illnesses [13]. 

Global recognition of the significance of zinc nutrition status in public health has expanded extremely in recent years, and zinc deficiency is now widely recognized as a leading risk factor for morbidity and mortality [6]. The connotations of zinc biology for human health are massive as about half of the world’s population is supposed to be at risk for zinc deficiency [14]. Zinc deficiency affects about 2.2 billion people around the world and has been ranked 11th among global risk factors for mortality and 12th for the burden of disease [13]. The main prevalence of zinc deficiency is observed in the developing countries of Africa, Asia, and Central America as well as in Andean countries [15]. In humans, the most common cause of zinc deficiency is reduced dietary intake, while other reasons include inadequate absorption, increased losses, or increased requirements [16]. 

Even though the relation between plasma zinc concentration (PZC) and clinical signs of zinc deficiency remains uncertain [17], clinical manifestations of zinc deficiency range from marginal to severe [18]. Wessells et al., 2016, proposed a cutoff using a PZC of 50 μg/dL (7.65 μmol/L) for severe zinc deficiency in adults [17]. Severe diseases are induced by impaired absorption or an iatrogenic-induced zinc-free diet. If unrecognized and not corrected, the zinc deficiency can turn fatal [18]. In addition, the nutritional status of children is an important part of the routine evaluation in pediatric consultations, not only in children displaying normal growth and development but also in those suffering from an acute or chronic illness. For this reason, it was hypothesized whether a deficient zinc status may be prevalent in a series of children with chronic diseases. Therefore, the main aim of this study was to evaluate zinc nutritional status and its relationship with nutritional indicators in a series of children with chronic diseases. Although serum zinc level in the whole series was normal, 69% of participants with a zinc intake lower than recommended might be at increased risk of zinc deficiency.

## 2. Materials and Methods

### 2.1. Study Site, Design, Participants, and Ethical Issues

This study was comparative and cross-sectional in design (Figure 1) and was carried out in the Nutrition Unit of the Pediatrics Service of the University Clinical Hospital in Valladolid, Spain. It was driven according to the Declaration of Helsinki, and the study protocol was accepted by the Ethical Committee of the University Clinical Hospital management board (INSALUD-Valladolid, 14 February 2002). Written informed consent was obtained from the relatives/guardians of all patients before taking part in this study. The number of cases seen in the Nutrition Unit during the eighteen months of the study determined the size of the sample. Eligible participants were selected by systematic sampling. The inclusion criteria were children under the age of 19 with a confirmed diagnosis of chronic disease. Chronic diseases include malnutrition of unknown cause, syndromic diseases, encephalopathies, kidney disease, hyperlipidemia, insulin-dependent diabetes mellitus, and eating disorders. Participants were classified by nutritional status (body mass index, BMI) into malnourished, obese, and eutrophic patients. Patients with cystic fibrosis [19] and acute infection and those who were hospitalized or refused to participate were excluded. The time of chronic diseases was shown in months.

### 2.2. Dietary Assessment

Participants were tutored to register all consumed foods and their amounts based on household measurements. Everything recorded was reviewed by a trained doctor during the interview. The reported daily intake of energy; protein; carbohydrates; lipids; monounsaturated, polyunsaturated, and saturated fats; fiber; vitamins; and minerals (including zinc) was calculated from the food consumption records of a 72 h prospective dietary survey (including one of weekend days), the week before the blood test. Nutrient sufficiency was assessed using percentage of Dietary Reference Intake (% DRI) or adequate intake using the Mataix Food and Health software, which provided the percentage of actual nutrient intakes with respect to Spanish recommendations [20,21]. Less than 80% DRI was the cutoff used to categorize a dietary intake as inadequate. In this series, no patient had taken micronutrient and vitamin supplements. The mean zinc intake fluctuated from 4.6 to 6.2 mg/day in children aged 1 to <3 years, from 5.5 to 9.3 mg/day in children aged 3 to <10 years, from 6.8 to 14.5 mg/day in children aged <10 to adolescents of 18 years, and from 8.0 to 14.0 mg/day in adults [22].

### 2.3. Assessment of Phenotypical Characteristics

Data on age and gender were collected using questionnaires. Anthropometric assessments of weight, height, wrist, hip, waist, and mid-arm circumference using standard techniques were performed. Weight and height were determined while wearing light clothing to the nearest 0.1 kg using a calibrated scale and a stadiometer to the nearest 0.1 cm, respectively. Body mass index (BMI) was calculated as weight/height^2^. Both obesity and undernutrition were estimated using BMI. The measure and Z-score of weight-for-age, height-for-age, weight-for-height, body mass index, BMI-height-age, and mid-arm muscle area, fat-free mass, and fat mass were calculated using Frisancho [23] and Orbegozo tables [24]. Triceps, biceps, and subscapular and suprailiac skinfolds were measured by standard methods with a Holtain Skinfold Caliper. Anthropometry and bioelectrical impedance analysis (BIA) (RJL Systems single frequency impedance analyzer) were used to measure body composition. Bone densitometry was measured by ultrasound (DBM Sonic 1200) through the bone conduction speed (BCS) of the last four fingers of the non-dominant hand [25].

### 2.4. Clinical Evaluation

During the evaluation of each patient, in addition to assessing the clinical and nutritional status, the presence or absence of diarrhea and some skin lesions related to zinc deficiency were evaluated [26], such as hyperpigmented skin, rough skin, keratosis/keratitis, dermatitis, bullous/pustular dermatitis, and alopecia.

### 2.5. Laboratory Methods

Three milliliters of blood from the cubital vein was collected from each of the study participants in fasting. The blood sample was immediately centrifuged for 10 min at 3000 rpm, wrapped in protective packaging, and then transported to the Laboratory of Instrumental Techniques of the Chemistry Department of the Valladolid University. Serum samples, previously stored at −18 °C, were slowly thawed, and then diluted (1:4) in deionized and demineralized water. Calibration curves (between 0 and 5 µg/dL) were made from aqueous solutions of the standards, using a wavelength of 213.9 nm, an analysis time of 4 s, an acetylene flow of 1.0 L/min, with a 0.5 nm slit, and a lamp intensity of 7.5 mA. The calibration was carried out in mg/L. All the material was previously washed with 20% nitric acid and washed with deionized water. Serum zinc was determined using an atomic absorption spectrophotometer (model PU9400 Philips) [27]. Hypozincemia was defined as serum zinc levels below 70 µg/dL in children under 10 years of age in both sexes and in females older than 10 years and <74 μg/dL in males older than 10 years [28,29]. Blood count, a complete biochemical analysis, and the activity of acute-phase proteins (inflammatory markers), including C-reactive protein (CRP) >4 U/L and erythrocyte sedimentation rata (ESR) in women >20 mm/h and men >15 mm/h, were measured using standardized methods. Serum prealbumin ≤18 mg/dL, albumin ≤3.5 g/dL as visceral protein reserve, transferrin ≤200 mg/dL, lymphocytes <2000 cell/mm^3^, total cholesterol (TC) >200 (mild-moderate risk) and >225 mg/dL (high risk), and low-density-lipoprotein cholesterol (LDL-C) >115 (mild-moderate risk) and >135 mg/dL (high risk), were used as cutoffs to evaluate abnormal values.

### 2.6. Statistical Analysis

A database was created to analyze the anthropometric, biochemical, and dietary results. The main variables studied were the serum zinc level and dietary zinc intake. Anthropometric, biochemical, body composition, and densitometry measurements were secondary. The distribution of anthropometric results (quantitatively and Z-scores) and biochemical data were described as mean, median, quartiles, standard deviation (SD) and range. Differences in nutritional parameters between gender, group of age, and normal and deficient were assessed using unpaired Student t-tests or Wilcoxon rank tests. Pearson’s correlation was used to test significant associations among variables, and the one-way analysis of variance (ANOVA test) was used to look for interactions in analytical values and gender, age group, and zinc status. Categorical data were evaluated by Pearson’s Chi-square test (X^2^) with Yates’s correction and Fisher’s exact test (FET). Simple and multiple linear regression analyses were calculated to study the significant associations between two and more meaningful correlations. The IBM SPSS software version 24.0 (IBM Corp., Armonk, NY, USA) was used to carry out the statistical analysis. The significance level was established at *p* < 0.05.

## 3. Results

Seventy-eight patients (43 females, 55%), 24 children with obesity, 30 with undernutrition, and 24 with normal BMI took part in this study. No one left the study. The mean age was 9.2 ± 4.8 years old with median 10 and range was 1–19 years. The mean time of illness was 65.8 ± 47 months (4 to 182 months), and there were no significant differences in time of illness between the three groups. Figure 2 shows the median, mean, and quartiles of the time of illness, serum zinc, and dietary zinc intake by nutritional status via BMI. The baseline characteristics in this series are shown in Table 1. In the whole series, the mean serum zinc was normal (86.8 µg/dL), including in all groups (87.4 µg/dL in the obese, 85.8 µg/dL in the undernourished, and 87.9 µg/dL in the eutrophic group). The mean zinc intake was normal in the obesity group (80.9% DRI) but was low in the undernutrition group (60.4% DRI) and in eutrophic patients (66.9% DRI). The mean dietary zinc intake was significant lower in the undernutrition group in comparison with that in the obesity group (*p* < 0.05), but the consumption in both groups no had significant differences with the eutrophic group. None of the five patients with hypozincemia had diarrhea, hyperpigmented skin, parakeratosis/keratitis, dermatitis, bullous/pustular dermatitis, or alopecia. Only one child with obesity and hypozincemia had rough skin. Of the 53 patients with low zinc intake, 11% showed rough skin (two malnourished, one eutrophic, three obese), 9% had dermatitis (four eutrophic and one obese), another 9% presented alopecia (three malnourished, one eutrophic, and one obese), and 2% presented hyperpigmented skin (one obese) and parakeratosis/keratitis (one malnourished). There was no significant association either between serum zinc and dietary zinc intake (Figure 3) or between them and the time of evolution of the disease (Figure 4).

In the whole series, 69% of participants had a dietary zinc intake lower than the recommendations and had deficiency of dietary zinc. The percentage of patients with deficient dietary zinc intake was higher in the undernutrition group (73%) and eutrophic group (71%) than in the obesity group (58%). There were no significant differences in the mean serum zinc and zinc intake by age and gender in all groups. There was no significant association between the time of illness and the level of serum zinc and zinc intake in the whole series and by groups of study. There were two cases with hypozincemia in the obese (8%) and undernourished (7%) groups and one case in the eutrophic group (4.2%). All patients with hypozincemia had dietary zinc deficiency. 

Table 2 shows the meaningful regression analysis among serum zinc, dietary zinc intake, and nutritional parameters studied in all groups. Lineal regression analysis revealed that, in the obesity group, serum iron and iron consumption had a significant association with serum zinc and zinc intake, respectively. In the undernutrition patients, there was a significant association between zinc intake and energy, carbohydrates, magnesium, iron, and niacin intake. In the eutrophic group, there was an association between serum zinc concentration and number of lymphocytes, and dietary zinc and niacin intake. Multiple regression analysis showed that only serum zinc in the obesity group had an association with wrist circumference and serum iron, while in the undernutrition group, carbohydrates and niacin consumption showed a significant relationship with dietary zinc intake.

## 4. Discussion

Nowadays, assessing zinc deficiency in any population is a challenging task because there is no gold standard measure to estimate zinc status [30]. Zinc is hard to evaluate effectively using laboratory tests due to its distribution throughout the body as a constituent of various proteins and nucleic acids. Unfortunately, there are no simple markers to assess marginal, mild, or moderate zinc deficiency in the individual [31]. However, it is recognized that serum/plasma zinc remains by far the most used method [30]. Even though blood test values are more significant than nutrient intake values with respect to the judgment of a patient’s nutritional status, serum zinc levels do not necessarily reflect zinc status at the cellular level due to strict homeostatic control mechanisms. In addition, the clinical effects of zinc deficiency may be present in the absence of abnormal laboratory indices [32]. Indirect indicators such as the prevalence of stunting or anemia and iron deficiency, as well as more direct indicators such as PZC are being used at present to estimate the prevalence of zinc deficiency in populations [33].

The actual prevalence of the problem in the Spanish population is poorly understood, and even less so in the child population. There are a few studies of zinc deficiency in children with chronic diseases [19]. In addition, as undernutrition has detrimental effects on the development of organs and growth [5], and childhood obesity and its comorbidities have also become a major health problem worldwide [7], both can be associated with zinc deficiency. Therefore, the main goal was to assess zinc nutritional status and its relationship with nutritional indicators in a series of children with chronic diseases. The results showed that even though serum zinc was normal and there were few cases with hypozincemia, the intake of zinc in the diet was significantly different in the three groups studied. Obese children had a better dietary zinc intake than the eutrophic and malnourished groups. In addition, most of children with lower zinc intake than recommendation might be at high risk of zinc deficiency. 

Across Europe, there is significant heterogeneity in the dietary reference values (DRVs) [34]. For zinc, the mean weighted correlations obtained by the EURRECA (EURopean micronutrient RECommendations Aligned) quality scoring system indicated that record retrieval and other validation methods had an acceptable correlation with the Food Frequency Questionnaire (FFQ) [35]. In this series, analyses of three-day food records’ results showed that the mean dietary zinc intake was significantly lower in the underweight (60% DRI) compared to that in the obesity group, which was normal (80% DRI, *p* = 0.040), but there were no significant differences with the eutrophic group (67% DRI), who also had low dietary zinc intake. It is important to note that the only group that had a normal mean zinc intake was the obesity group. The mean dietary zinc intake in the three groups was significantly lower than the National Health and Nutrition Examination Survey (NHANES) II study’s control group (178% RDA, *p* < 0.001) [36]. However, the mean value in the obesity group (12 mg/day) was the only that was high and differs significantly from the Anthropometry, Intake, and Energy Balance in Spain (ANIBES) study (8.1 mg/day, *p* = 0.001) [37], compared with the undernutrition (9 mg/day) and eutrophic (10 mg/day) groups, whose means correspond to the ANIBES study. 

On the other hand, in this study, 65.6% of children under 9 years old, 56.5% between 9 and 12 years old, and 86.4% of adolescents between 13 and 17 years old had a higher deficient zinc intake in contrast to 31% of children between 9 to 12 years and 65% of adolescents between 13 to 17 years in the ANIBES study [37]. In addition, it is striking that the undernutrition group (73.3%) and the control group (71%) had the highest percentages of cases with deficient zinc intake compared to the obesity group (58%). The results in the ANIVA (Anthropometry and Child Nutrition of Valencia) study (73.5%) were comparable to those of the undernutrition and eutrophic groups in this study [38]. Olza et al. showed that 83% of the Spanish inhabitants included in the ANIBES study, whereas in this study (68.8%), did not meet the European recommendations (EFSA) for zinc [37]. It is estimated that up to 17.3% of the world’s population is at risk of inadequate zinc intake [16], and countries with a prevalence of poor dietary zinc intake >25% are considered at high risk of zinc deficiency [6]. Therefore, because in this study, all groups had more than 25% prevalence of low zinc intake, they would be at high risk of zinc deficiency [28,39]. 

Surprisingly, the daily intake for the entire series was hyperproteic (276% DRI), with high consumption of cholesterol (266% DRI), slightly low intake of carb (79.5% DRI), and normal total lipid intake (111% DRI). Although the diet was hyperproteic for all groups and there was a direct and meaningful association between protein and zinc intake in the whole series (*p* = 0.432, *r* = 0.000) and in the undernutrition group (308% DRI, *p* = 0.649, *r* = 0.000), as reported by Wapnir [40], it seems that the amount of dietary protein intake was not enough to provide adequate zinc intake for all of them. This fact could be because of the type of protein ingested [37]. The protein content of the ingested food has a positive effect on zinc absorption due to the release of amino acids and peptides upon degradation [41]. According to Olza et al., 2017, in the Spanish ANIBES study, the principal sources of zinc for the whole population were meat and meat products (28.5%), cereals and grains (25.5%), and milk and dairy products (15.8%). Fish (5.7%), vegetables (5.2%), and ready-to-eat meals (4.8%) complete the list of food with which more than 85% of the total intake of zinc is reached [37]. Furthermore, Shakur et al. pointed out in Bangladeshi children that PZC predicted protein-energy malnutrition [42].

As known, inadequate dietary intake of absorbable zinc is likely the primary cause of zinc deficiency in most situations [43], and what is more important, human zinc absorption is substantially higher in the presence of protein from animal sources than with plant-based protein [44]. Although many grain- and plant-based foods are still good sources of zinc, the bioavailability of zinc from grains and plant foods is much lower than that from animal foods [45]. Moreover, inhibitors of zinc absorption are probably the most important contributing factors in zinc malabsorption [46]. This can be explained by the fact that soil zinc deficiency has been associated with human zinc deficiency in developing countries. In addition, although there is moderate zinc deficiency in soils in Spain, zinc deficiency in the Spanish population is generally defined as low risk [47]. Perhaps, the reason why 69% of participants had low zinc intake is that the international recommendations for zinc intake are not accurate and could be too high.

Plasma/serum zinc concentrations have been reported to decline with age [48]. Nevertheless, in the three groups, a significant decrease in serum zinc level with age was not seen. Serum zinc levels in the children of this study may have depended on zinc intake in the week prior to the blood test. Mean serum zinc concentration was normal in the three groups, obese (87 µg/dL), undernutrition groups (85 µg/dL), and the eutrophic group (88 µg/dL). There were no significant differences between them and with the NHANES 2011–2014 study (82.7 μg/dL) [49]. However, the mean serum zinc of the three differs significantly from the Navarra study carried out in 3887 children between 4 and 17 years (112.8 µg/dL, *p* = 0.000) [50] and from the Ortega et al.’s study performed on 357 Spanish schoolchildren between 8 and 13 years (101.7 µg/dL, *p* = 0.000) [51]. In the present study, there were few cases with hypozincemia. Only two cases presented hypozincemia in the obesity group (8%) and in the malnutrition group (7%), and only one case in the eutrophic group (4.2%). None of them had symptoms of severe zinc deficiency (<50 µg/dL) [17]. This result is in agreement with the prevalence of low serum zinc concentration in the U.S., in which, it was esteemed that around 4% of children and 8% adults had hypozincemia [52]. 

It is important to note that this fact raises the question of how humans maintain an adequate level of serum zinc despite having a poor zinc intake. In this study, although 69% of the children had deficient zinc in the diet, the mean serum zinc level was normal, and only a few patients simultaneously showed hypozincemia with low zinc intake. In addition, no patient with hypozincemia had symptoms of severe zinc deficiency, but some children with low zinc intake had symptoms that could be associated with zinc deficiency. This situation is due to the homeostatic mechanisms that maintain the serum zinc levels, and no notable changes are observed even though zinc intake is restricted [53]. In case of risk of very low or marginal zinc intakes for a long period of time, intestinal absorption increases between 59% and 84% [54], and intestinal losses are reduced, due to better intestinal preservation linked to stimulation of the synthesis of membrane transporters and by a decrease in pancreatic and biliary secretion. In addition, kidney losses decrease [55]. Nevertheless, according to Lowe et al., the PZC may fall in response to factors unrelated to zinc status or dietary zinc intakes, such as infection, inflammation, exercise, stress, or trauma. Postprandial PZCs have been reported to fall up to 19%. On the contrary, tissue catabolism during inanition can release zinc into the circulation, producing a transient increase in circulating zinc levels [56]. These facts add to the lack of a reliable and sensitive marker of zinc status and the non-specific nature of the disease’s symptoms associated with suboptimal zinc intake [34].

Interestingly, as shown in Figure 4, there was no significant association between the time of illness and the level of serum zinc and zinc intake for the entire series as in the three groups. In other words, the time of evolution of the patients’ diseases would not be an important factor in the serum level of zinc and in its intake. This study is the first to establish this type of relationship. Furthermore, there was no association between serum zinc level and zinc intake. Three meta-analyses reached the same conclusion that serum zinc levels are not related to zinc intake [30,57,58]. Nevertheless, according to Hess, serum zinc level responds noticeably and fast (within less than 2 weeks in some cases) to severe dietary zinc restriction. There is undoubted evidence of a strong connection between dietary zinc intake and serum zinc level under these experimental conditions. In contrast, similar experimental data are lacking for people who had been chronically exposed to marginal zinc intakes prior to any dietary interventions [41].

Additionally, there were different associations between nutritional parameters, dietary zinc intake, and serum zinc (Table 2). Linear regression analysis revealed that in the obesity group, serum zinc and dietary zinc intake had showed a direct association with serum iron and iron intake, respectively. Nevertheless, in the undernutrition group, there was only a direct association between dietary zinc and iron intakes. As known, zinc status is related to iron metabolism among human subjects [59]. Although zinc and iron deficiency often coexist, growing evidence suggests that serum zinc may affect hemoglobin levels regardless of iron status [60]. Furthermore, when dietary Fe:Zn index is greater than 2:1, zinc absorption will be inhibited because the Fe carrier, transferrin, which also carries zinc, becomes saturated [61]. In addition, systemic iron deficiency and low iron levels are seen in obesity and closely associated with adiposity [62].

It is interesting to find that lineal regression analysis showed a meaningful association between serum zinc and bicipital skinfold Z-score (negative correlation) in the undernutrition group. This result agrees with Perrone et al.’s study, who found a negative correlation between serum zinc and the sum of triceps and subscapular skinfold thickness [63]. In contrast, serum zinc had a significant association (negative correlation) with wrist circumference (WrC) in the obesity group. WrC is a simple and easy-to perform anthropometric index, which is not subject to measurement errors, it can be used both in screening and clinical assessment procedures for obesity-related metabolic consequences [64] and cardiometabolic risk in children and adolescents [65]. This result may suggest that WrC could be a new functional indicator to study zinc status and may support the need to use parameters other than BMI. For example, the fat free mass by BIA (r = −0.491, *p* = 0.024) and BCS (r = −0.453, *p* = 0.039) had a negative and significant association with dietary zinc intake in this group.

Results show that there was a meaningful association between serum zinc and magnesium intake in the obesity group, and dietary zinc and magnesium intake in the undernutrition group. According to Suliburska et al., lower magnesium status was proved in obese children and adolescents. Magnesium plays a major role in regulating insulin effect and insulin-mediated glucose uptake in the body; and a deficit of this element disturbs carbohydrate metabolism [66]. Quite the reverse, Nielsen noted that marginal zinc deficiency significantly decreased magnesium excretion and increased magnesium concentration in the bone. These findings suggest a possible interaction between zinc and magnesium. Furthermore, it is known that magnesium deficiency leads to a reduction in bone mass, abnormal bone growth, and an increase in skeletal fragility in animal models [67].

Undernutrition is the result of an imbalance between the supply of nutrients/energy and the demand of the body to guarantee its functions and growth [5]. The positive correlation between dietary zinc and protein intake may mean that efficient zinc absorption depends on dietary intake and protein digestion [68]. Gibson et al. found that there was a positive correlation between zinc, energy, and other nutrients intake (protein, fat, carbohydrate, fiber, calcium, and iron) [28]. Zinc is essential for lean body mass synthesis, and its deficiency was reported to increase the energy cost of tissue deposition [69]. Furthermore, in the undernutrition and eutrophic groups, there was a significant association between intake of zinc and niacin. Niacin is a precursor of NAD+ (Nicotinamide Adenine Dinucleotide), the substrate for the activity of DNA repair enzyme PARP-1 (poly(ADP-ribose) polymerase 1) and, consequently, may contribute to maintaining genomic stability [70]. In a study conducted in 14 alcoholic pellagra patients (seven male control subjects), the effect of zinc supplementation suggested that zinc interacts with niacin metabolism in these patients through a probable mediation by vitamin B-6 [71].

Additionally, there were two important associations that are worth highlighting. On the one hand, dietary zinc intake had a significant association with prealbumin in the obesity group. Prealbumin is the precursor to albumin. In the NHANES 2011–2014 study, a low serum zinc level was related to lower serum albumin concentrations [49]. In a controlled metabolic study, when the zinc intake was reduced from 16.5 to 5.5 mg/day, but the dietary protein was unchanged, urinary urea nitrogen, serum prealbumin, albumin, and retinol-bonding protein all decreased significantly [72]. On the other hand, in the eutrophic group there was a significant association between serum zinc and lymphocytes. Zinc is critical for the normal development and function of cells that mediate both innate and adaptive immunity [73]. Concerning adaptive immune function, zinc deficiency affects lymphocyte number and function [74]. T lymphocytes are particularly vulnerable to zinc deficiency [75]. Furthermore, not only 24% (19/78 cases) of the patients had a high ESR, but also all the children with hypozincemia (TEF, *p* = 0.001). The immune system is especially sensitive to changes in the level of zinc; in fact, it appears that each response is related in some way directly or indirectly to zinc [76], and these changes may act as a stimulus and interfere with both specific and non-specific immunity [77].

Another point to consider is how to improve the lower dietary zinc intake in the diet observed in this study. Moran et al. (EURRECA) found that an adult with a zinc intake of 14 mg/day has a zinc serum/plasma concentration 6% higher than an individual with a zinc intake of 7 mg/day [57]. A child with a zinc consumption of 14 mg/day has a zinc serum/plasma level 9% higher than a child whose intake is 7 mg/day. These data provide an estimate of the dose–response relationship between zinc intake and serum/plasma zinc concentration required to achieve normal or optimal levels of zinc status [30] and reduce the gap between dietary zinc deficiency and hypozincemia. Furthermore, without affecting zinc intake and as a result of dietary intervention in obese adolescents, a redistribution of zinc is observed due to a decrease in body fat [78] and an increase of fat-free mass in children with chronic kidney disease [79].

This is the first study on nutritional zinc status in a series of children with chronic disease. This study indicates that although 69% of the children were zinc deficient in the diet; the mean serum zinc level was normal and only a few cases showed hypozincemia with low zinc intake simultaneously. In addition, results of this study tell us about a high risk of zinc deficiency not only in the group of patients with obesity (58%) and undernutrition (73%) but also in the eutrophic group (70%). These results should make us reflect on the idea that a state of zinc deficiency can occur even in children with adequate nutritional status and not only in patients with malnutrition, both obesity and undernutrition. In this type of patient, an intervention strategy must be carried out, as an assessment of the quality/quantity of zinc intake and its bioavailability in relation to other nutrients [80].

This study makes it clear that there is insufficient information on zinc status in healthy children but even less in children with chronic diseases. Although in this study, the serum zinc levels could only indicate the zinc status of these patients, the intake of zinc in the diet helps us to know the high risk of zinc deficiency in patients with a lower zinc intake than recommended. It is vital to know whether children are really taking enough amounts of zinc. A likely response to this disturbing problem would be to reconsider zinc intake recommendations on the one hand and promote zinc status assessments in the context of national health and nutrition surveys on the other.

Considering all the highlights, these findings support the theoretical framework of this study. However, this study has the limitation of the sample size, while its fortes include determining zinc nutritional status and the association between serum zinc levels and dietary zinc intake with other nutritional parameters. This study raises more questions, and the results support the need for a larger study to improve zinc awareness in these patients and also to establish the necessary quantity required in case of its supplementation. We should consider the diagnosis of zinc nutritional status in primary health prevention since its routine evaluation in hospitals would allow its early detection and its treatment would reduce morbidity.

## 5. Conclusions

In this study of children with chronic disease, the mean serum zinc level was normal with some cases of hypozincemia. Only the obese group had an adequate mean dietary zinc intake, with 58.3%, 73%, and 71% low zinc intake in the obesity, malnutrition, and eutrophic groups, respectively. In the entire series, 69% of participants with deficient dietary zinc intake might be at an elevated risk of zinc deficiency.

## Figures and Tables

**Figure 1 nutrients-13-01121-f001:**
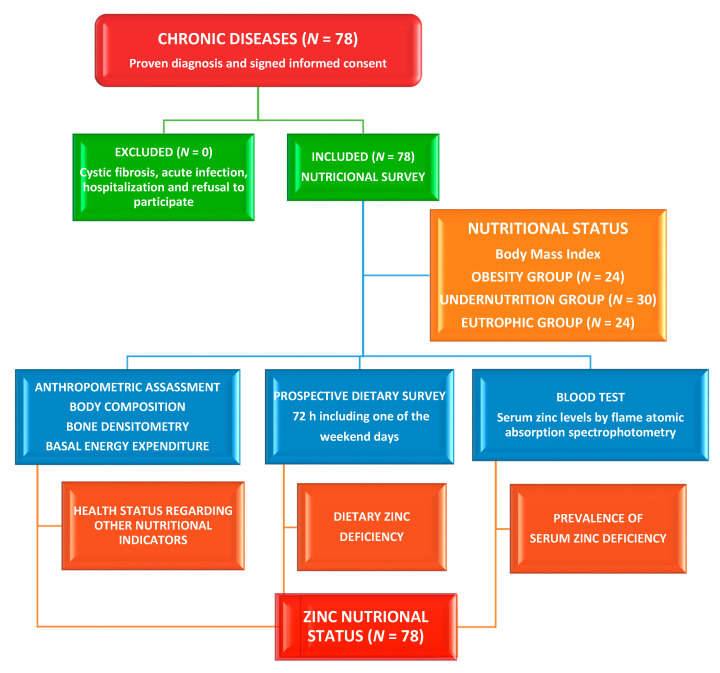
Flow diagram assignment of patients with chronic diseases (*N* = 78).

**Figure 2 nutrients-13-01121-f002:**
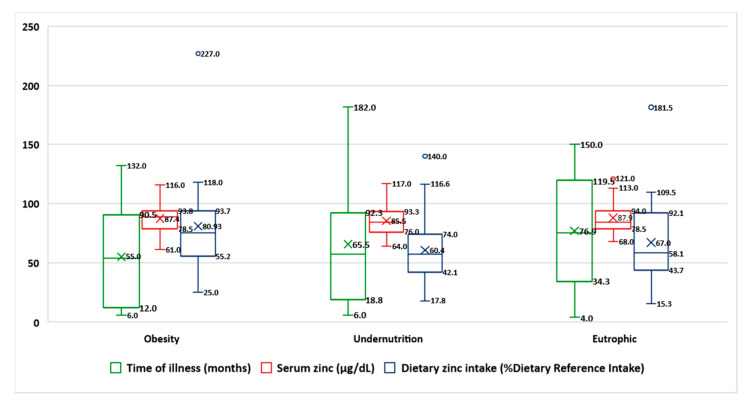
Median, mean, and quartiles of the time of illness, serum zinc, and dietary zinc intake in the whole series by nutritional status via body mass index (*N* = 78).

**Figure 3 nutrients-13-01121-f003:**
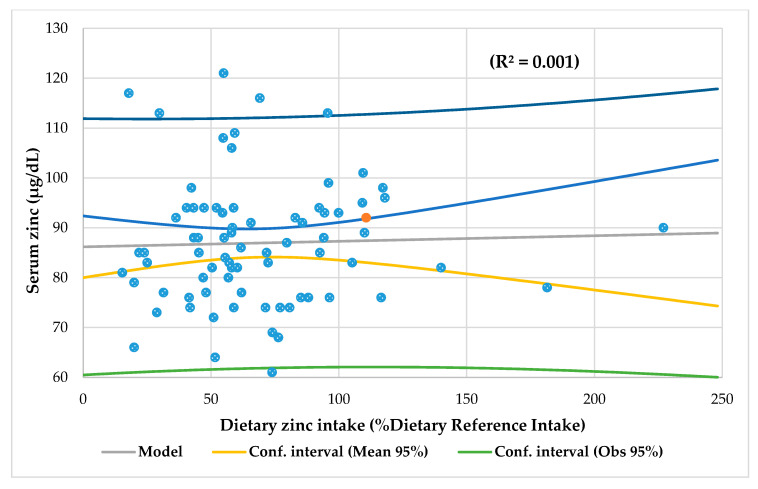
Correlation between serum zinc and dietary zinc intake in the entire series (*N* = 78).

**Figure 4 nutrients-13-01121-f004:**
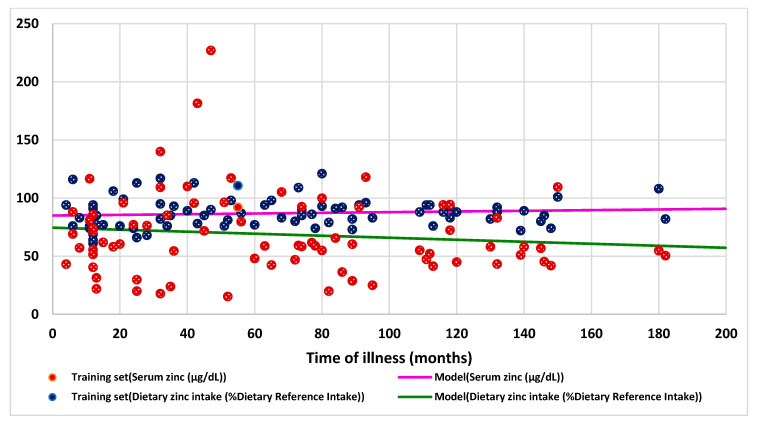
Correlation between serum zinc, dietary zinc intake, and the time of illness in the entire series (*N* = 78).

**Table 1 nutrients-13-01121-t001:** Baseline characteristics of the children with chronic disease (*N* = 78 *).

Characteristics	Obesity (*N* = 24)	Undernutrition (*N* = 30)	Eutrophic (*N* = 24)	*p-*Value
	Mean ± SD	Mean ± SD	Mean ± SD	
Female (%)	15 (62.5)	17 (56.7)	11 (45.8)	0.472
Age (years)	11.3 ± 3.9	7.2 ± 4.9	10.5 ± 4.7	0.240
Children (age in years)	6.9 ± 2.8	4.5 ± 2.8 *	7.2 ± 3.3	0.026 ***
Adolescent (age in years)	13.4 ± 2.1	13.5 ± 1.8	14.5 ± 2.3	0.363
Time of chronic disease (months)	55 ± 38.4	65.5 ± 52.8	76.9 ± 47.5	0.282
Wrist circumference (cm)	16.6 ± 3.3	11.6 ± 2.4	14 ± 3.3	0.000 ***
Biceps skinfold Z-score	2 ± 1.9	−1.9 ± 0.6	−0.7 ± 1.2	0.000 ***
Blood Analytic				
Prealbumin (NV > 18 mg/dL)	23.7 ± 5.6	20.9 ± 5.8	21 ± 5.8	0.185
Zinc (NV 70–120 µg/dL)	87.4 ± 12.2	85.8 ± 12.6	87.9 ± 13	0.761
Iron (NV 60–100 µg/dL)	80.7 ± 20.7	75.6 ± 39.9	80.4 ± 28	0.806
Lymphocytes (NV > 2000 cell/mm^3^)	2887 ± 1527	3605 ± 1641	2615 ± 719	0.028 ***
Hypozincemia cases (%)	2 (8.3)	2 (6.7)	1 (4.2)	
Prospective Dietary Survey				
Dietary zinc (NV 80–120% DRI)	80.9 ± 40	60.4 ± 30	66.9 ± 35	0.110
Dietary zinc intake (mg/day) **	12 ± 6.1	9 ± 4.5	10 ± 5.2	0.109
Carbohydrates (NV 80–120% DRI)	81.3 ± 50	79.6 ±25.9	77.5 ± 27.4	0.182
Energy (NV 80–120% DRI)	87 ± 22.7	95.6 ± 25.5	97.8 ± 23.2	0.276
Niacin (NV 80–120% DRI)	154 ± 49.3	126 ± 56	142 ± 50.3	0.161
Magnesium (NV 80–120% DRI)	107 ± 41	103.5 ± 41.1	103.7 ± 34.5	0.942
Iron (NV 80–120% DRI)	157 ± 127	181 ± 118	205 ± 145.6	0.531
Dietary zinc deficiency cases (%)	14 (58.3)	22 (73.3)	17 (70.8)	0.388

Abbreviations: % DRI: percentage of dietary reference intake. NV: normal values. * 78 chronic disease patients were selected, included, and analyzed. No patients refused to participate. ** Zinc intake: NV 4.6 to 6.2 mg/day in children aged 1 to <3 years, from 5.5 to 9.3 mg/day in children aged 3 to <10 years, from 6.8 to 14.5 mg/day in children aged <10 year to adolescents of 18 years, and from 8.0 to 14.0 mg/day in adults [22]. *** *p*-value < 0.05.

**Table 2 nutrients-13-01121-t002:** Regression serum zinc, dietary zinc intake by nutritional parameters in both groups (*N* = 78).

	Obesity (*N* = 24)	Undernutrition (*N* = 30)	Eutrophic (*N* = 24)
Serum ZincConcentration	Dietary ZincIntake	Serum ZincConcentration	Dietary ZincIntake	Serum ZincConcentration	Dietary ZincIntake
Linear regression analysis	Wrist circumference*r* = 0.229,*p* = 0.018Magnesium intake*r* = 0.244,*p* = 0.0014Serum iron*r* = 0.228,*p* = 0.021	Iron intake*r* = 0.366, *p* = 0.002Prealbumin*r* = 0.303, *p* = 0.010	Biceps skinfoldZ-score*r* = 0.260,*p* = 0.008	Energy intake*p* = 0.278, *p* = 0.003Carbohydrates intake*r* = 0.331, *p* = 0.001Niacin intake*r* = 0.254, *p* = 0.005Magnesium intake*r* = 0.442, *p* = 0.000Iron intake(*r* = 0.222, *p* = 0.002	Lymphocytes*r* = 0.269,*p* = 0.009	Niacin intake*r* = 0.203,*p* = 0.027
Multiple regression analysis	Wrist circumference andSerum iron*r* = 0.487,*p* = 0.008			Carbohydrates and Magnesium intake*r* = 0.572, *p* = 0.009

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
