# Peer review of "Zinc Nutritional Status in a Series of Children with Chronic Diseases: A Cross-Sectional Study"

_nutrients, 2021, doi:10.3390/nu13041121_

Round 1

Reviewer 1 Report

Line 51: Clinical signs of Zn deficiency are mentioned, but completely missing in RESULTS. - Line 76: "Chronical diseases" are mentioned however not listed. Line 84: Informed consent was given by children ???Line 120: Description of sample collection/measurement is insufficient: What was used: serum or plasma? Precision data of Zn determination are missing. Line 68: Zn deficiency ranges from marginal to severe: insert the cut-offs! - Line 124: Electrolyte concentrations should be presented in umol/L. The chosen cut-off (10.71 umol/L) is very low (compare Hotz et al: Am J Clin Nutr 2003; 78: 756-64). - Dietary Zn and serum concentrations should correlate: Please demonstrate (prove) this fact (at length!) at least in the few cases of hypozincemia! - Most important: What is the message for the reader??On what parameter shall he rely? E.g., must the choice of food be improved - or the food questionnaire and related conditions??

Author Response

Changes introduced in the manuscript nutrients-1144441

We acknowledge reviewer # 1 for their useful comments, which helped to improve the quality of the manuscript.

  • Comment 1. Line 51: Clinical signs of Zn deficiency are mentioned, but completely missing in RESULTS

Author’s Response: Thanks for your comment. Information on clinical signs of zinc deficiency is added to material and methods (Line 137), and the results (Line 192) as follows:

(Line 137-141) “2.4. Clinical evaluation

During the evaluation of each patient, in addition to assessing the clinical and nutritional status, the presence or absence of diarrhea and some skin lesions related to zinc deficiency were evaluated [26], such as hyperpigmented skin, rough skin, keratosis/keratitis, dermatitis, bullous/pustular dermatitis and alopecia.”

(Line 192-198) “None of the 5 patients with hypozincemia had diarrhea, hyperpigmented skin, parakeratosis/keratitis, dermatitis, bullous/pustular dermatitis, nor alopecia. Only one child with obesity and hypozincemia had rough skin. Of the 53 patients with low zinc intake, 11% (2 malnourished, 1 eutrophic, 3 obese) showed rough skin, 9% (4 eutrophic and 1 obese) had dermatitis and alopecia (3 malnourished, 1 eutrophic and 1 obese) and 2% presented hyperpigmented skin (1 obese) and parakeratosis/keratitis (1 malnourished).”

  • Comment 2. Line 68: Zn deficiency ranges from marginal to severe: insert the cut-offs!

Author’s Response: Thank you for your comment. Suggested information has been added as follows:

Even though the relation between plasma zinc concentration (PZC) and clinical signs of zinc deficiency remains uncertain [17], the clinical manifestations of zinc deficiency ranges from marginal to severe [18]. Wessells et al. (2016) proposed a cut-off for severe zinc deficiency in adults, using a PZC of 50 μg/dL (7.65 μmol/L) [17].” (Line 71-74).

  • Comment 3. Line 76: "Chronic diseases" are mentioned however not listed. Line 84: Informed consent was given by children???

Author’s Response: Thanks for your comment. The list of chronic diseases is mentioned and listed as follows:

Chronic diseases include patients with malnutrition of unknown cause, syndromic dis-eases, encephalopathies, kidney disease, hyperlipidemia, insulin-dependent diabetes mellitus, and eating disorders.” (Line 98-100).

  • Comment 4. Line 84: Informed consent was given by children???

Author’s Response: Thank you for your comment. This sentence has been changed as follows:

“Written informed consent was obtained from the relatives/guardians of all patients before taking part in this study.” (Line 93-94).

  • Comment 5. Line 120: Description of sample collection/measurement is insufficient: What was used: serum or plasma? Precision data of Zn determination are missing.

Author’s Response: Thank you for your comment. The description of the sample collection/measurement and the precision data for the determination of serum zinc has been improved. As explained below, serum was used:

“The blood sample was immediately centrifuged for 10 min at 3000 rpm, wrapped in protective packaging, and then transported to the Laboratory of Instrumental Techniques of the Chemistry Department of the Valladolid University. Serum samples, previously stored at −18 ° C, were slowly thawed, then diluted (1: 4) in deionized and demineralized water. Calibration curves (between 0 - 5 µg/dL) were made from aqueous solutions of the standards, using a wavelength of 213.9 nm, an analysis time of 4 seconds, an acetylene flow of 1.0 L/min, with a 0.5 nm slit and a lamp intensity of 7.5 mA. The calibration was carried out in mg/L. All the material was previously washed with 20% nitric acid and rinsed with deionized water. Serum zinc was determined using an atomic absorption spectrophotometer (model PU9400 Philips) [27].” (Line 144-155).

  • Comment 6. Line 124: Electrolyte concentrations should be presented in umol/L. The chosen cut-off (10.71 μmol/L) is very low (compare Hotz et al: Am J Clin Nutr 2003; 78: 756-64).

Author’s Response: Thank you for your comment. In this study, the authors prefer to present the results in µg/dL as presented by the author that you have suggested we use as cut-off points for serum zinc levels, as follows:

Hipozincemia was defined as serum zinc levels below 70 µg/dL in children <10 years of both sexes and in women aged ≥10 and <74 μg/dL in men aged ≥10 [28-29].” (Line 156-157).

  • Comment 7a. Dietary Zn and serum concentrations should correlate: Please demonstrate (prove) this fact (at length!) at least in the few cases of hypozincemia!

Author’s Response: Thank you for your comment. As mentioned below, there is no strong evidence of a correlation between dietary zinc and serum concentrations. It is important to note that despite the low zinc intake from the diet, there were only a few cases of hypozincemia:

It is important to note that this fact raises the question of how humans maintain an adequate level of serum zinc despite having a poor zinc intake. In this study, although 69% of the children were zinc deficient in the diet, the mean serum zinc level was normal and only a few patients were hypozincemia with low zinc intake at a time. In addition, no patient with hypozincemia had symptoms of severe zinc deficiency, but some children with low zinc intake had symptoms that could be associated with zinc deficiency. This situation is due to the homeostatic mechanisms that maintain the serum zinc levels, and no notable changes are observed despite the fact that zinc intake is restricted [53]. In case of risk of very low or marginal zinc intakes for a long period of time, intestinal absorption increases between 59-84% [54], and intestinal losses are reduced, due to better intestinal preservation linked to stimulation of the synthesis of membrane transporters and by a decrease in pancreatic and biliary secretion. In addition, kidney losses decrease [55]. Nevertheless, according to Lowe et al., plasma zinc concentration can fall in response to factors unrelated to zinc status or dietary zinc intakes, such as infection, inflammation, exercise, stress, or trauma. Postprandial plasma zinc concentrations have been reported to fall up to 19%. Conversely, tissue catabolism during starvation can release zinc into the circulation, causing a transient increase in circulating zinc levels [56]. These facts add to the lack of a reliable and sensitive marker of zinc status and the non-specific nature of the dis-ease’s symptoms associated with sub-optimal zinc intake [34].

Interestingly, as it is shown in Figure 4, for the entire series as in the three groups, there was no significant association between the time of illness and the level of serum zinc and zinc intake. In other words, the time of evolution of the patients' diseases would be an important factor neither in the serum level of zinc nor in its intake. This study is the first to establish this type of relationship. Furthermore, there was no association between serum zinc level and zinc intake. Three meta-analyses reached the same conclusion that serum zinc concentrations are not related to zinc intake [30, 57-58]. Nevertheless, according to Hess, serum zinc concentration responds appreciably and fairly rapidly (within less than 2 weeks in some cases) to severe dietary zinc restriction. There is convincing evidence of a strong relationship between dietary zinc intake and serum zinc concentration under these experimental conditions. In contrast, similar experimental data are lacking for individuals who had been chronically exposed to marginal zinc intakes before any dietary interventions [41].” (Line 336-368).

  • Comment 7b. Most important: What is the message for the reader??On what parameter shall he rely on? E.g., must the choice of food be improved - or the food questionnaire and related conditions??

Author's Response: Thank you for your comment. Below, we underscore the message of this study to the reader, and the challenges we face regarding this elusive and essential micronutrient:

This is the first study on nutritional zinc status in a series of children with chronic disease. This study indicates that despite 69% of the children was zinc deficient in the diet; the mean serum zinc level was normal and only had a few cases were hypozincemia with low zinc intake simultaneously.  In addition, results in this study tell us about a high risk of zinc deficiency not only in the group of patients with obesity (58%) and undernutrition (73%) but even in the eutrophic group (70%). These results should make us reflect on the idea that even a state of zinc deficiency can occur in children with adequate nutritional status and not only in patients with malnutrition, both obesity and undernutrition as shown in this study. In this type of patient, an intervention strategy must be carried out, either in the form of an assessment of the quality/quantity of zinc intake and its bioavailability in relation to other nutrients [78].

This study makes it clear that there is insufficient information on zinc status in healthy children and even less in children with chronic diseases. Although in this study, the serum zinc levels alone could indicate the zinc status of these patients, the intake of zinc in the diet helps us to know the high risk of zinc deficiency in those patients with a lower zinc intake than recommended. A likely response to this disturbing problem would be to reconsider zinc intake recommendations on the one hand and promote zinc status assessments in the context of national health and nutrition surveys on the other.” (443-462).

Thank you for your contribution, we really appreciate the review made that enriches the presentation of this research work.

Reviewer 2 Report

Dear authors,

Thank you very much for your interesting and detailed investigation about nutritional zinc status in children with chronic diseases.

I just have minor points of revision:

Line 77

  • Subjects with a lower zinc intake than the recommendations do not necessarily have a zinc deficiency. Please use a different kind of wording. Suggestion: “participants with a zinc intake lower the recommendations”
  • Please change the wording throughout the text, e.g. line 355 and 359.
  • Please indicate that values of the blood analytic are more meaningful than values of the nutrient intake regarding judgment of the nutritional statues of a patient.

Line 84

  • Please mention the name and location of your hospital.

Line 88

  • Please name and list the chronic diseases your patients presented. It might be of interest to see if there are diseases influencing the resorption and metabolism of zinc.

Figure 1

  • Please correct the title to: “Flow diagram of patient assignment”
  • Please add number of patients in each box (n=?)
  • Lines from leading from one box to others are confusing.
  • Wording is confusing. Please do not use the same wording for different contents (nutritional status: One time you are talking about BMI and another about nutrient intake)
  • Please order boxes in a way that the reader can recognize, in which way patient groups were built. (I tried to do figure for you, but the program does not do what I want to show you, sorry!)
  • If you want to give an overview of the different investigations, please use a separate chart.

Line 101

Please mention which DRI you used. Did you use Spanish recommendations or Europeans or the ones of the WHO or other ones?

Table 1

  • Do not print “Female” bold.
  • Erase the line under “Female”
  • Add the units after “children, adolescents, time…., wrist…, dietary…, carbo.., energy…….
  • Add reference values to the blood analytic
  • Please add protein- and fat-intake as well, especially you are taking about the protein intake in the discussion in line 220.
  • Do you have an explanation why the carbohydrate intake is only about 40% of the recommendations, even in the group with the obese patients?

Figure 2

  • Why do you give figure 2 since you do not explain or talk about it in the text?
  • If you want to explain and show it, please add units, recommendations and reference values.
  • Boxplots do not present the mean, but the median. Please change the wording.

Author Response

Changes introduced in the manuscript nutrients-1144441

We acknowledge reviewer # 2 for their useful comments, which helped to improve the quality of the manuscript.

Thank you very much for your interesting and detailed investigation about nutritional zinc status in children with chronic diseases. I just have minor points of revision:

  • Comment 1. Line 77: Subjects with a lower zinc intake than the recommendations do not necessarily have a zinc deficiency. Please use a different kind of wording. Suggestion: “participants with a zinc intake lower the recommendations”

Author’s Response: Thank you for your comment. This recommendation has been followed throughout the manuscript, for example, as follows:

The 69% of participants showed dietary a zinc intake lower deficiency than recommended and might be at an elevated risk of zinc deficiency.” (Line 28-29)

  • Comment 2. Line 355: Please change the wording throughout the text, e.g. line 355 and 359.

Author’s Response: Thank you for your comment. It was changed as follows:

Only the obese group had an adequate mean dietary zinc intake, with 58.3%, 73%, and 71% low zinc intake in the obesity, malnutrition, and eutrophic groups, respectively. This 69% of participants with deficient dietary zinc intake might be at an elevated risk of zinc deficiency.” (Line 474-481).

  • Comment 3. Please indicate that values of the blood analytic are more meaningful than values of the nutrient intake regarding judgment of the nutritional status of a patient.

Author’s Response: Thank you for your comment. It has been added as follow:

Nowadays, assessing zinc deficiency in any population is a challenging task because there is no gold standard measure to estimate zinc status [30]. Zinc is difficult to measure adequately using laboratory tests due to its distribution throughout the body as a component of various proteins and nucleic acids. Regrettably, there are no simple markers of marginal, mild or moderate zinc deficiency in individual [31] However, it is recognized that serum/plasma zinc stays by far the most used method [30]. Even though blood test values are more significant than nutrient intake values with respect to the judgment of a patient's nutritional status, serum zinc levels do not necessarily reflect cellular zinc status due to tight homeostatic control mechanisms. In addition, clinical effects of zinc deficiency can be present in the absence of abnormal laboratory indices [32]. Indirect indicators such as the prevalence of stunting or anemia, iron deficiency, as well as more direct indicators such as plasma zinc concentrations are being used at present to estimate the prevalence of zinc deficiency in populations [33].” (Line 237-249).

  • Comment 4. Line 84: Please mention the name and location of your hospital.

Author’s Response: Thank you for your comment. The name and location of the hospital have been added as follows:

The study was comparative and cross-sectional in design (Figure 1) carried out in the Nutrition Unit of the Pediatrics Service of the University Clinical Hospital in Valladolid, Spain”  (Line 89-90).

  • Comment 5. Line 88: Please name and list the chronic diseases your patients presented. It might be of interest to see if there are diseases influencing the resorption and metabolism of zinc.

Author’s Response: Thank you for your comment. The list of chronic diseases is mentioned and listed as follows:

Chronic diseases include patients with malnutrition of unknown cause, syndromic dis-eases, encephalopathies, kidney disease, hyperlipidemia, insulin-dependent diabetes mellitus, and eating disorders.” (Line 98-100).

This point is very interesting, however cystic fibrosis has been added as an exclusion criterion since the results obtained in relation to zinc have already been published:

Cystic fibrosis patients [19], acute infection, hospitalization, and refusal to participate were exclusion criteria.” (Line 101-102).

  • Comment 6. Figure 1: Please correct the title to: “Flow diagram of patient assignment”

Author’s Response: Thank you for your comment. It was changed as follows:

“Flow diagram of patients assignment with chronic diseases.” (Line 103).

  • Comment 7. Figure 1: Please add number of patients in each box (n=?)

Author’s Response: Thank you for your comment. The number was added.

  • Comment 8. Figure 1: Lines from leading from one box to others are confusing.

Author’s Response: Thank you for your comment. It was changed.

  • Comment 9. Figure 1: Wording is confusing. Please do not use the same wording for different contents (nutritional status: One time you are talking about BMI and another about nutrient intake)

Author’s Response: Thank you for your comment. Figure 1 was changed.

  • Comment 10. Figure 1: Please order boxes in a way that the reader can recognize, in which way patient groups were built. (I tried to do figure for you, but the program does not do what I want to show you, sorry!)

Author’s Response: Thank you for your comment and for trying to do it. The figure was changed.

  • Comment 11. Figure 1: If you want to give an overview of the different investigations, please use a separate chart.

Author’s Response: Thank you for your comment, the authors prefer not to give an overview of the different investigations.

  • Comment 12. Line 101: Please mention which DRI you used. Did you use Spanish recommendations or Europeans or the ones of the WHO or other ones?

Author’s Response: Thank you for your comment. It was added as follows:

“…which provided the percentage of actual nutrient intakes with respect to Spanish recommendations [20-21].” (Line 115-116).

  • Comment 13. Table 1: Do not print “Female” bold. Erase the line under “Female”

Author’s Response: Thank you for your comment. The changes were done.

  • Comment 14. Table 1: Add the units after “children, adolescents, time…., wrist…, dietary…, carbo.., energy…….

Author’s Response: Thank you for your comment. The units were added.

  • Comment 15. Table 1: Add reference values to the blood analytic

Author’s Response: Thank you for your comment. References to the blood test were added in Table 1, and in the material and methods as follows:

“Blood count, a complete biochemical analysis and the activity of acute-phase proteins as inflammatory markers, including C-reactive protein (CRP) > 4 U/L and ESR woman >20 mm/h, men >15 mm/h were assessed by standardized methods. Serum prealbumin ≤18 mg/dL, albumin ≤3.5 g/dL as visceral protein reserve, transferrin ≤ 200 mg/dL, lymphocytes < 2000 cell/mm3, total cholesterol (TC) >200 (mild-moderate risk) and >225 mg/dL (high risk), and low-density lipoprotein-cholesterol (LDL-C) >115 (mild-moderate risk) and >135 mg/dL (high risk), were used as cutoff to evaluate abnormal values.” (Line 158-165).

  • Comment 16. Line 220: Please add protein- and fat-intake as well, especially you are taking about the protein intake in the discussion in line 220.

Author’s Response: Thank you for your comment. Protein and fat intake have been added as follows:

Surprisingly, the daily intake for the entire series was hyperproteic (276 ± 176% DRI), with high consumption of cholesterol (266 ± 131% DRI), slightly low intake of carb (79.5 ± 35% DRI) and normal total lipid intake (111 ± 40% DRI). Although for all groups, the diet was hyperproteic and there was a direct and significant association between protein and zinc intake in the whole series (p = 0.432, r = 0.000) and in the undernutrition group (308 ± 222%DRI, p = 0.649, r = 0.000), as it was reported by Wapnir [40], the amount of dietary protein intake seems that it was not enough to provide adequate zinc intake for the all of them. This fact could be because of the type of protein ingested [37]. The protein content of the consumed food has a positive effect on zinc absorption due to the release of amino acids and peptides upon degradation [41].” (Line 292-302).

  • Comment 17. Do you have an explanation why the carbohydrate intake is only about 40% of the recommendations, even in the group with the obese patients?

Author’s Response: Thank you for your comment, there was an error writing the data and it has been changed.

  • Comment 18. Figure 2: Why do you give figure 2 since you do not explain or talk about it in the text? If you want to explain and show it, please add units, recommendations and reference values. Boxplots do not present the mean, but the median. Please change the wording.
  • Author’s Response: Thank you for your comment. In figure 2, units have been added and the results are mentioned according to the median, mean and quartiles, and it has also been referenced in the text as follows:

“Figure 2 shows median, mean and quartiles of the time illness, serum zinc and dietary zinc intake by nutritional status via BMI.” (Line 183-184)

Thank you for your contribution, we really appreciate the review made that enriches the presentation of this research work.

Round 2

Reviewer 1 Report

Unfortunately, the study raises more questions than answers are given. Zinc seems to be important - but do children really take (in-)sufficient amounts?

Author Response

Changes introduced in the manuscript nutrients-1144441

We acknowledge reviewer # 1 for their useful comments, which helped to improve the quality of the manuscript.

  • Comment. Unfortunately, the study raises more questions than answers are given. Zinc seems to be important - but do children really take (in-)sufficient amounts?

Author’s Response: Thank you for your comment, we agree with them and we have included them as follows:

“This study makes it clear that there is insufficient information on zinc status in healthy children but even less in children with chronic diseases. Although in this study, the serum zinc levels only could indicate the zinc status of these patients, the intake of zinc in the diet helps us to know the high risk of zinc deficiency in patients with a lower zinc intake than recommended. It is vital to know whether children are really taking enough amounts of zinc. A likely response to this disturbing problem would be to recon-sider zinc intake recommendations on the one hand and promote zinc status assessments in the context of national health and nutrition surveys on the other.

Considering all the highlights, these findings support the theoretical framework of this study. However, a limitation of the study is the small sample size, while its strengths include determining zinc nutritional status and the association between serum zinc levels and dietary zinc intake with other nutritional parameters. This study raises more questions, and the results support the need for a larger study to improve understanding of zinc in these patients, but also to determine the necessary amount required in case of its supplementation. We should consider the diagnosis of zinc nutritional status in primary health prevention since its routine evaluation in hospitals would allow its early detection and its treatment would reduce morbidity.” (Line 500-516).

Thank you for your contribution, we really appreciate the review made that enriches the presentation of this research work.
